# DALD: Improving Logits-based Detector without Logits from Black-box LLMs

**Cong Zeng**[1*]   **Shengkun Tang**[1*]   **Xianjun Yang**[2]   **Yuanzhou Chen**[3]
**Yiyou Sun**[4]   **Zhiqiang Xu**[1]   **Yao Li**[5]   **Haifeng Chen**[4]   **Wei Cheng**[✉4]   **Dongkuan Xu**[6]
MBZUAI[1]   University of California, Santa Barbara[2]
University of California, Los Angeles[3]   NEC Labs America[4]
University of North Carolina, Chapel Hill[5]   NC State University[6]
{cong.zeng, shengkun.tang, zhiqiang.xu}@mbzuai.ac.ae   xianjunyang@ucsb.edu
{sunyiyou, haifengchen, weicheng}@nec-labs.com   dxu27@ncsu.edu
yuanzhouchen@cs.ucla.edu   yaoli@ad.unc.edu

## Abstract

The advent of Large Language Models (LLMs) has revolutionized text generation, producing outputs that closely mimic human writing. This blurring of lines between machine- and human-written text presents new challenges in distinguishing one from the other – a task further complicated by the frequent updates and closed nature of leading proprietary LLMs. Traditional logits-based detection methods leverage surrogate models for identifying LLM-generated content when the exact logits are unavailable from black-box LLMs. However, these methods grapple with the misalignment between the distributions of the surrogate and the often undisclosed target models, leading to performance degradation, particularly with the introduction of new, closed-source models. Furthermore, while current methodologies are generally effective when the source model is identified, they falter in scenarios where the model version remains unknown, or the test set comprises outputs from various source models. To address these limitations, we present **D**istribution-**A**ligned **L**LMs **D**etection (`DALD`), an innovative framework that redefines the state-of-the-art performance in black-box text detection even without logits from source LLMs. `DALD` is designed to align the surrogate model's distribution with that of unknown target LLMs, ensuring enhanced detection capability and resilience against rapid model iterations with minimal training investment. By leveraging corpus samples from publicly accessible outputs of advanced models such as ChatGPT, GPT-4, and Claude-3, `DALD` fine-tunes surrogate models to synchronize with unknown source model distributions effectively. Our approach performs SOTA in black-box settings on different advanced closed-source and open-source models. The versatility of our method enriches widely adopted zero-shot detection frameworks (DetectGPT, DNA-GPT, Fast-DetectGPT) with a *plug-and-play* enhancement feature. Extensive experiments validate that our methodology reliably secures high detection precision for LLM-generated text and effectively detects text from diverse model origins through a singular detector. Our method is also robust under the revised text attack and non-English texts.

## 1   Introduction

Large language models (LLMs) such as ChatGPT[1], GPT-4[2], Llama[3–5] and Claude-3[6] have profoundly impacted both industrial and academic domains, reshaping productivity across various

---

*Equal contribution. The code and data are released at https://github.com/cong-zeng/DALD

38th Conference on Neural Information Processing Systems (NeurIPS 2024).

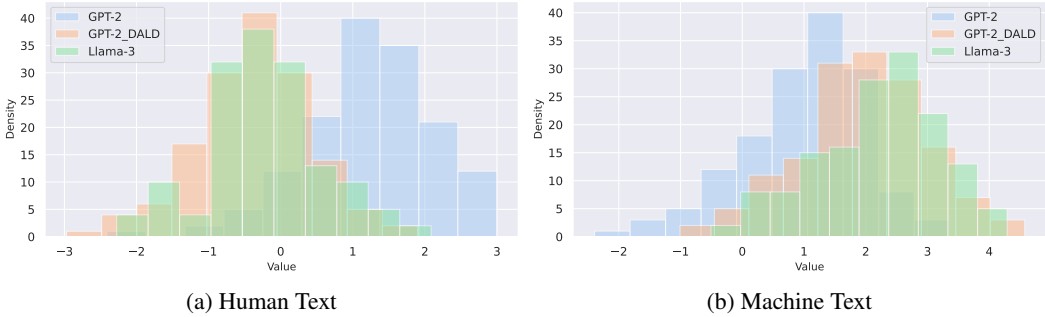

(a) Human Text            (b) Machine Text

Figure 1: The probability curvatures distribution of the surrogate model (GPT-2), the target model (Llama-3) and the model after alignment (GPT-2_DALD) on human-written passages and machine-generated passages from PubMed dataset.

sectors including news reporting, story writing, and academic research[7]. Nevertheless, their misuse also raises concerns, particularly regarding the dissemination of fake news[8], the proliferation of malicious product reviews[9], and instances of plagiarism[10]. Instances of AI-synthesized scientific abstracts deluding scientists[11, 12] have raised doubts about the reliability of scientific discourse. Accurate and reliable machine-generated text detection methods are necessary in order to address these issues[13–17].

Methods for detecting text generated by Large Language Models are broadly categorized into watermarking[18–21], training-based classifiers[22–26], and zero-shot detectors. Watermarking methods discreetly embed identifiable markers within the text output, striving to retain the model's linguistic integrity. However, this tactic is implementable solely by the model provider. Training-based classifiers, while effective, are costly and often lack the agility to adapt to new domains or model updates. Our emphasis is on zero-shot detectors that exploit the intrinsic differences between text written by machines and humans, offering the advantage of being generally training-free.

Most zero-shot detectors primarily depend on analyzing model output logits for detection. Notably, DetectGPT[27] operates on probability divergence based upon principles of perturbation theory, while DNA-GPT[28] harnesses reprompting-based probability divergence, and Fast-DetectGPT[29] builds on variations in conditional probability distributions. In scenarios requiring the scrutiny of black-box models, these strategies commonly leverage a surrogate model to approximate the behavior of the target model. However, this approach is doubly flawed: firstly, detection efficacy is inextricably linked to a meticulously tailored surrogate model, with different surrogate models often necessary for accurate detection across various proprietary LLMs; secondly, the fleeting nature of LLM updates renders past surrogates, once effective, obsolete against new versions. For instance, our analysis of the performance of Fast-DetectGPT[29], using `GPT-Neo-2.7B` as a surrogate, against freshly updated closed-source models reveals erratic and predominantly diminishing accuracy, as contextualized in Figure 2, with particularly stark declines in performance on iterations like `GPT-3.5-1106`, highlighting the intrinsic limitation of static surrogate models in adapting to LLM progressions.

In our study, we seek to address the following pivotal inquiries: 1) Can we devise a feasible, cost-effective strategy to refine the probability distribution similarity between the surrogate model and opaque black-box LLMs? 2) Does enhancing the alignment of the surrogate model's probability distribution with that of the target black-box LLM improve detection outcomes for current logits-based detection methods? 3) Is it attainable to develop a universal detection model capable of adapting swiftly to updates across various target LLMs? Addressing the first question, our findings, as demonstrated in Figure 1,

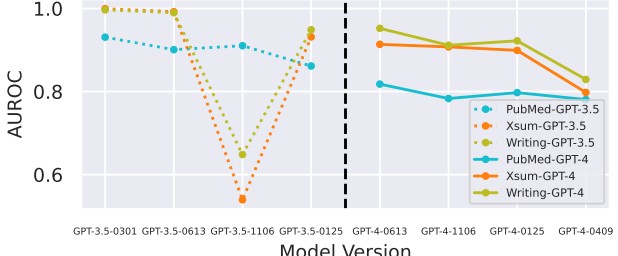

Figure 2: The performance comparison of a static surrogate model on different target models including ChatGPT (GPT-3.5) and GPT-4. The results are based on Fast-DetectGPT with `GPT-Neo-2.7B` as the surrogate model.

reveal that our optimized surrogate model (GPT-2_DALD) mirrors the distribution of the target model more closely, in contrast to the original surrogate model's significantly divergent distribution from the target model.

In this paper, we introduce an innovative yet straightforward distribution-aligned framework for black-box LLM detection, dubbed DALD (**D**istribution-**A**ligned **LLM** **D**etection). Our methodology focuses on synchronizing the surrogate model's distribution with the proprietary target model's distribution. Concretely, we accumulate a compact dataset (<10K samples) from the publicly shared outputs of leading models and subsequently fine-tune our surrogate model using this dataset to better approximate the target model's distribution. We provide the theoretical analysis of the surrogate model distribution alignment in **Appendix 6**. Our methodology builds upon the following observation:

> *In logits-based detection methods, a surrogate model that closely mirrors the probability distribution curves of the target black-box LLM is instrumental in enhancing detection accuracy.*

We posit that this observed effect stems from the foundational assumptions inherent in logits-based detectors and proceed to examine the ramifications of this postulate in tackling the third question.

To sum up, our contributions are as follows:

- The introduction of DALD, a framework that significantly improves the performance of surrogate models in detecting LLM-generated text generated by both closed-source and open-source models.
- DALD's unique ability to enhance detection without reliance on knowledge of the source model – a game-changer in a domain where the source is often unknown.
- The capability of a single detector, enabled by DALD, to accurately identify text from varying sources, democratizing detection across diverse LLM outputs.
- DALD's agility in keeping pace with rapid updates of LLMs, ensuring the latest models fall within its detection capabilities without extensive retraining.

## 2 Related Work

**Detection of LLMs-Generated Text.**  The burgeoning capabilities of advanced large language models (LLMs) underscore the imperative for robust methodologies aimed at detecting these models. Specifically, the detection is to distinguish whether a given text originates from a language model on the condition that the model is known (White-box)[27] or unknown (Black-box)[30]. The earlier work focused on feature-based methods, like[31–34]. While in the era of LLMs, the training-based methods[31, 35–37] are aroused to counter with LLMs's strong ability to produce high-quality text. They usually involve training a binary classifier using text generated by AI or humans. Besides, zero-shot detectors leverage the inherent statistical feature differences between LLMs and human-generated text without requiring training, including probability curvature (DetectGPT[27]), N-gram divergence (DNA-GPT[28]), and conditional probability curvature (Fast-DetectGPT[29]), the editing distance of the output[38], and style representations[39], enhancing their ability to adapt to new data distributions and source models.

**Black-box Detection.**  Given the proprietary nature of the latest LLMs[1, 2, 6], there is a critical need for effective black-box detection methods. Present techniques falter when direct access to the source model is restricted. The training-based methods, like OpenAI text classifier[40], GPTZero[22], G3detector[41], and GPT-Sentinel[42] usually closely adhere to the specific distributions of text domains and source models during training, thereby lacking generalization ability and robustness on model updates. For zero-shot methods [43, 38, 27–29, 44, 45, 34, 46, 14, 47–49] in the black-box detection settings, they usually rely on a surrogate model for scoring. However, the efficacy of these surrogate models often falls short compared to white-box detection, where access to the source model is available. Moreover, these detection frameworks suffer from diminished accuracy when language models undergo updates[50], which intrinsically evolve through exposure to varied datasets and human input[30]. This study presents an innovative black-box detection method for LLM-generated text, greatly enhancing surrogate model performance while adeptly accommodating the rapid evolution of LLMs. Our approach diverges from conventional training-intensive techniques by requiring only a minimal dataset for effective training.

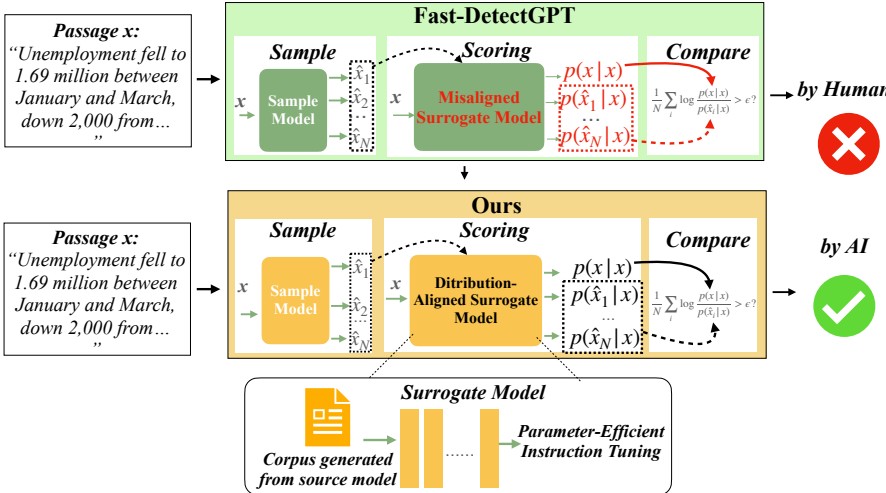

Figure 3: An overview of our proposed DALD framework. Our method aligns the distribution of the surrogate model and the target model.

## 3 Method

### 3.1 Task and Settings

Our task is to detect whether the given input passage $x = [x_1, ..., x_L]$ ($L$ is the sequence length) is produced by an AI model $f_{\text{tar}}$ or a human, which can be considered as a binary classification task. Typically, there are two different task settings for LLM detection, namely white-box and black-box detection. In the black-box setting, we only have access to the generated text, treating the language model as a "black box" where we input text and receive output without knowing the internal workings or probabilities. In the white-box setting, we have additional information about the model, specifically the output probabilities $p(x_l|x_{[1:l-1]})$ for each token at each position $l$ in the text. However, in the practical scenario, it is usually difficult to get access to the source model, especially widespread but closed-source models such as ChatGPT, GPT-4 and Claude-3. Therefore, we focus on improving the black-box detection without any access to the source model logits in our setting.

### 3.2 Logits-based Detection Methods

Logits-based LLM detection methods compute a metric by discrepancy gap hypothesis of humans and machines for classification. For example, based on the observation that the LLM-generated text occupies negative curvature regions of the model's log probability function, DetectGPT[27] proposes to utilize the source model for scoring, which refers to the white-box settings. Following DetectGPT, Fast-DetectGPT[29] replaces the perturbations-based sampling method with conditional probability sampling to accelerate the inference speed and improve the detection performance. Formally, given an input passage $x$ and the target source model $p_\theta$, Fast-DetectGPT chooses another accessible but open-sourced model $s_\theta$ for scoring, which is called the surrogate model. Together with a sampling model $q_\varphi$, Fast-DetectGPT defines the conditional probability $p(\widehat{x}|x)$ as

$$p(\widehat{x}|x) = \prod_l s_\theta(\widehat{x}_l|x_{<l}), \tag{1}$$

where $\widehat{x}$ is a sample generated by the sampling model $q_\varphi$. The detection process typically consists of a three-stage procedure. The sampling step uses a sampling model to generate alternative samples $\widehat{x}$ conditioned on $x$ based on the next token prediction. Following the sampling step, the process proceeds by calculating the conditional score. The conditional probability can be obtained through a single forward pass of the scoring model, utilizing $x$ as the input. All the conditional probabilities of samples can be obtained in the same predictive distribution. Finally, compare conditional probabilities of the passage and samples to calculate the curvature.

**The Challenge.** In black-box settings, selecting the appropriate surrogate model in black-box settings is crucial for achieving accurate and reliable results since there is distribution misalignment

between the surrogate model and the target model. Poorly chosen surrogate models may lead to bad results and a lack of explainability. Besides, with the closed-source trend of newly published LLM models, the performance can drop significantly when applied to new and advanced models, which can limit their utility and effectiveness. How to obtain a surrogate model that can fit the target models, especially closed-source models is a challenging task in black-box settings.

### 3.3 Distribution-Aligned Black-Box Detection

**Misalignment of Surrogate Model and Target Model.** Our method is motivated by the observation that there is a distribution gap between the given surrogate model and the target source model as shown in Figure 1. The goal of our method is to obtain a surrogate model to approach the distribution of the target model by utilizing the texts generated by the target model. To achieve that, we propose a novel and simple framework to train a distribution-aligned surrogate model, which outperforms SOTA black-box methods with a small-size dataset (<10K). The architecture of our method is shown in Figure 3. Our framework consists of two steps in total. The first step is to collect small-size training data generated by the source model from the publicly shared outputs. With the training dataset, we finetune the surrogate model to align the distribution of the source model.

**Alignment Data Collection.** Given the target model $f_{tar}$ and surrogate model $f_{sur}$, in order to align the distribution of the surrogate model and target model, we collect a small-size dataset $\mathcal{S} = \{(P_i, X_i)\}_{i=1}^N$ for a specific target model, referred as alignment dataset, where $N$ refers to the number of collected samples, $P_i$ is the text for prompting and $X_i$ is the corpus generated by the target model $f_{tar}$. The model version of the collected data should be exactly the same as the target model, especially for closed-source models such as ChatGPT and GPT-4. For example, if the test data is generated by `GPT-4-0613`, then all of the texts in the dataset $\mathcal{S}$ should also be generated by `GPT-4-0613`. We utilize the collected dataset $\mathcal{S}$ to finetune the surrogate model $f_{sur}$ to align the distribution with target model $f_{tar}$.

**Distribution-Aligned Surrogate Model Training.** As illustrated in Figure 3, our approach expands the scoring step of previous logits-based methods such as Fast-DetectGPT by incorporating an additional surrogate model finetuning step. Given the surrogate model $f_{sur}$, we construct the Low-Rank Adaptation (LoRA)[51] of surrogate model $f_{sur}$ for faster and more stable fine-tuning. The LoRA model $f_{sur+\Theta}$ is trained with a collected dataset while the parameter of the original surrogate model $f_{sur}$ is frozen. With collected dataset $\mathcal{S} = \{(P_i, X_i)\}_{i=1}^{K_1}$ where $K_1$ is the number of samples, we concatenate the prompt and generated text as $y = [P, X]$. The model $f_{sur+\Theta}$ utilizes the tokenized $x$ as input and is trained in a self-supervised learning manner. The training objective of our fine-tuning is:

$$\max_{\Theta} \sum_{y=[P,X] \in \mathcal{S}} \sum_{l=l(P)+1}^{l(P)+l(X)} \log p(y_l | y_{<l}; sur + \Theta), \tag{2}$$

where $l(X)$ denotes the length of a passage $X$, and $y_l$ is the next token to be predicted. In order to disable the influence of the prompt, we follow typical instruction tuning to mask the gradient of the prompt. As shown in Figure 1, after training, the misaligned model generates a similar distribution as the target source model, demonstrating the effectiveness of our method. Following that, the distribution-aligned surrogate model can be utilized to compute the logits for downstream decisions.

Under an assumption on the sample complexity of fine-tuning with the above loss function, we theoretically demonstrate the effectiveness of fine-tuning on approximating the target model in the following theorem, using **conditional probability curvature** from Fast-DetectGPT [29]:

**Theorem 1.** With fine-tuning sample size $K_1 = \Omega(\text{poly}(\Delta/L))$, with probability $1 - \delta$, we have that given a text segment $X$ with length $l$, the conditional probability curvature between the two models is bounded by

$$\left| \mathbf{d}(X, f_{\text{sur}}) - \mathbf{d}(X, f_{\text{tar}}) \right| \leq \Delta/3.$$

A detailed proof of this theorem can be found in Appendix 6. Given the hypothesis that there is a positive gap $\Delta$ in conditional probability curvature between human-generated text and machine-generated text, the corresponding gap calculated from the surrogate model will still be significant.

Table 1: Detection accuracy comparison on three source models ChatGPT (`GPT-3.5-Turbo-0301`), GPT-4 (`GPT-4-0613`) and Claude-3 (`claude-3-opus-20240229`). Our method surpasses previous methods on all passages generated from different source models.

| Method | ChatGPT | GPT-4 | | | Claude-3 | | |
|---|---|---|---|---|---|---|---|
| | PubMed | PubMed | XSum | Writing | PubMed | XSum | Writing |
| RoBERTa-base | 0.6298 | 0.5327 | 0.7475 | 0.5186 | 0.4961 | 0.8564 | 0.6707 |
| RoBERTa-large | 0.7168 | 0.5898 | 0.6830 | 0.3800 | 0.5334 | 0.7888 | 0.5178 |
| Likelihood | 0.8924 | 0.8103 | 0.8096 | 0.8528 | 0.8543 | 0.9383 | 0.9542 |
| Entropy | 0.2877 | 0.3036 | 0.4451 | 0.3545 | 0.2940 | 0.3856 | 0.1844 |
| LogRank | 0.8847 | 0.7996 | 0.8041 | 0.8303 | 0.8481 | 0.9420 | 0.9437 |
| LRR | 0.7793 | 0.6860 | 0.7405 | 0.7212 | 0.7468 | 0.8989 | 0.8761 |
| NPR | 0.6917 | 0.5950 | 0.6726 | 0.8192 | 0.6610 | 0.8584 | 0.9167 |
| Detect-GPT | 0.6626 | 0.5806 | 0.6940 | 0.8270 | 0.6562 | 0.8652 | 0.9232 |
| DNA-GPT | 0.7788 | 0.7171 | 0.7100 | 0.7849 | 0.7442 | 0.9410 | 0.9471 |
| Fast-DetectGPT | 0.9309 | 0.8179 | 0.9136 | 0.9521 | 0.8900 | 0.9828 | 0.9445 |
| DALD (Ours) | **0.9853** | **0.9785** | **0.9954** | **0.9980** | **0.9630** | **0.9867** | **0.9981** |

## 4 Experiments

### 4.1 Setups

**Datasets & Evaluation Metric.** We follow Fast-DetectGPT using four datasets in the black-box detection evaluation, including Xsum[52], WritingPrompts[53], WMT-2016[54] and PubMedQA[55]. We randomly sample 150 examples of each dataset as human-written texts. Then based on the samples, we prompt the target closed-source models by API to generate the corresponding texts using the 30 tokens of human-written text as the machine-generated text. For text diversity and quality, we employ a temperature of 0.8 which is the same setting in Fast-DetectGPT. For the training dataset, we collect the corpus from the publicly shared outputs of leading models. Following previous works[29], we compute the accuracy in the area under the receiver operating characteristic (AUROC) to evaluate the performance of all methods. We also provide the area under the precision and recall (AUPR) in Appendix 8.6.

**Source & Surrogate Models.** To validate our idea in black-box detection, we include the most advanced closed-source LLMs from OpenAI: ChatGPT, GPT-4, and Claude-3 from Anthropic. Since these models keep being updated by their owner company, we use the version `GPT-3.5-turbo-0301` for ChatGPT, `GPT-4-0613` for GPT-4, and `claude-3-opus-20240229` for Claude-3 if not specified. We utilize `Llama2-7B` as the surrogate model in Table 1. Note that our method can be adapted to any open-source model. Therefore, we provide our results on other surrogate models in Table 3.

**Baseline Methods.** We consider training-based baselines and zero-shot baselines. We mainly consider three strong baselines for black-box detection: Detect-GPT[27], DNA-GPT[28] and Fast-DetectGPT[29]. Detect-GPT. Detect-GPT applies `T5-3B`[56] as a sampling model to generate perturbed texts and utilizes `GPT-Neo-2.7B`[57] as a surrogate model to compute the probability curvature of perturbed texts. After that, perturbation discrepancy is obtained to determine whether the given text is generated by AI or humans. Fast-DetectGPT uses `GPT-J-6B`[58] and `GPT-Neo-2.7B` as the sampling model and surrogate model respectively to compute the conditional probability curvature. Finally, DNA-GPT utilizes `GPT-Neo-2.7B` as a surrogate model to regenerate the texts for metric computation. Details about other baselines are described in Appendix 7.

**Implementation Details.** We collect the data of ChatGPT and GPT-4 from WildChat[59] while the data of Claude-3 is generated by calling Claude-3 using the prompts from WildChat. During training, we randomly choose 5K prompts and responses. We applied the instruction tuning to model training to ignore the human-written prompts. For surrogate model training, we apply parameter-efficient fine-tuning (PEFT) via Low-Rank Adaptation (LoRA). We do not tune the hyperparameters carefully. Therefore, more details about training parameters can be found in the Appendix 7. For training time, our method finetunes `Llama-2-7B` with 5K samples on 4 A6000.

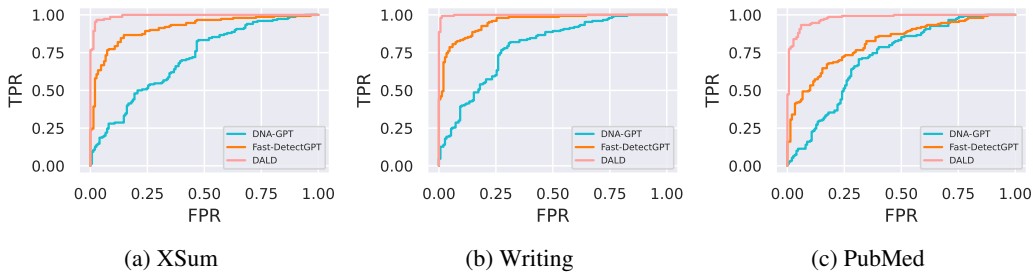

(a) XSum               (b) Writing             (c) PubMed

Figure 4: The FPR-TPR curve of different methods on XSum, Writing and PubMed dataset. The results show that our method achieves highest score at low FPR compared with DNA-GPT and Fast-DetectGPT.

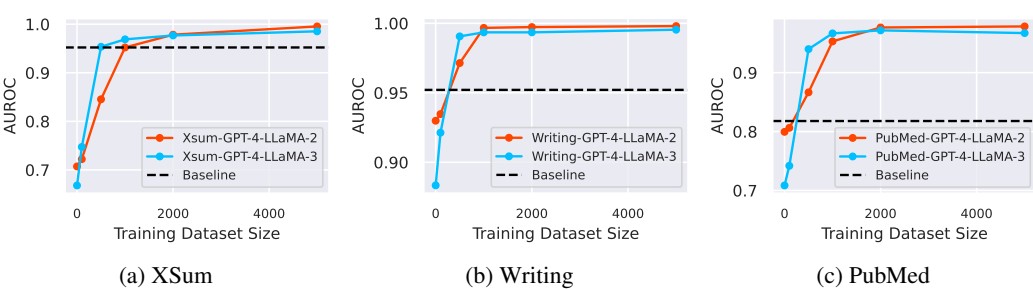

(a) XSum               (b) Writing             (c) PubMed

Figure 5: AUORC results from our fine-tuned surrogate model with different training dataset size.

## 4.2  Main Results

**Black-Box Machine-Generated Text Detection.**   We compare our method with mainstream black-box LLM detection methods in Table 1. The detection accuracy shows that our method achieves the best performance compared with other methods including Detect-GPT, DNA-GPT and Fast-DetectGPT. Moreover, it is noteworthy that our method obtains more than 99% AUROC on XSum-GPT-4, Writing-GPT-4 and Writing-Claude-3. As shown in Figure 4, the ROC curve for DALD achieves the highest TPR at the same FPR across all three datasets compared with DNA-GPT and Fast-DetectGPT, indicating superior performance. Besides, comparing the performance across different datasets in previous methods, we can observe that the Writing dataset is much easier to detect while PubMed is the hardest. The possible reason is that PubMed is a much more medical-specific dataset while the related corpus is not comprised of surrogate model pre-training. However, our method gains significant improvement upon Fast-DetectGPT. For example, on PubMed-GPT-4, Fast-DetectGPT only obtains 0.8179 AUROC while our method achieves 0.9785, demonstrating the effectiveness of distribution alignment in our method.

## 4.3  Experimental Analysis

**Dataset Size.**   Our method requires only a small amount of data for training. The model converges quickly with the small-size dataset. To show the training efficiency of our method, we provide the results with various training dataset sizes in Figure 5. For each dataset, we use GPT-4 as the target source model while `Llama-2-7B` and `Llama-3-8B` as the surrogate model respectively. With the increasing amount of training data, the performance increases rapidly and stays stable with more data. Moreover, with around 500 (up to 1000) training samples, the performance of our method matches the baseline on both Llama-2 and Llama-3. Besides, our method achieves its best performance and exceeds the baseline with around 2000 training data, which indicates that with only a little training effort, our method can achieve incredibly better performance than the baseline, even more than 99%.

Table 2: The results comparison of our method trained with the combination of different data sources. Our method achieves comparable results with more data sources, demonstrating the generalizability of our method and potentially leading to training a universal surrogate model for all closed-source models. †: Train the surrogate model separately to each test set. *: Train one surrogate model for all test sets.

| Method | ChatGPT PubMed | GPT-4 PubMed | XSum | Writing | Claude-3 PubMed | XSum | Writing |
|---|---|---|---|---|---|---|---|
| Baseline | 0.9051 | 0.7995 | 0.7072 | 0.9299 | 0.8877 | 0.9143 | 0.9248 |
| DALD $^\dagger$(1 source) | **0.9853** | 0.9785 | 0.9954 | 0.9980 | **0.9942** | 0.9994 | **0.9993** |
| DALD *(1 source) | 0.9829 | 0.9785 | 0.9954 | 0.9980 | 0.9875 | 0.9993 | 0.9977 |
| DALD *(2 sources) | 0.9832 | 0.9803 | **0.9981** | **0.9986** | 0.9875 | 0.9994 | 0.9976 |
| DALD *(3 sources) | 0.9827 | **0.9809** | 0.9968 | 0.9985 | 0.9864 | **0.9996** | 0.9982 |

**Generalizability.** Our method aligns with the distribution between the surrogate model and source model with the texts generated by source models. In this section, we further explore the generalizability of our method where we follow several settings: 1) train the surrogate model separately using the data from the corresponding source model, 2) train the model with the single data source and evaluate it on unknown source models (one-for-all), 3) train the model with mixed data sources. We conduct a group of experiments following the settings, as shown in Table 2. We use `Llama2-7B` as the surrogate model and a total 5K training data in all settings. The dataset with a single data source includes 5K texts from the corresponding source model in the first setting and only GPT-4 in the one-for-all setting while two data sources consist of 2.5K ChatGPT and GPT-4 texts, respectively. Three data sources refer to the combination of 1.3K texts each from ChatGPT, GPT-4 and Claude-3. Surprisingly, the models trained with more data sources achieve better accuracy on GPT-4. However, the performance on PubMed-ChatGPT only shows negligible degradation in one-for-all and mixed data source settings. The superior performance in the one-for-all setting implies the surrogate model trained with `DALD` can be extended to texts of unknown source models. The results demonstrate the generalizability of our method, leading to training a universal surrogate model for all closed-source models and detecting the machine-generated texts without knowing the model source. Finally, the results in the one-for-all setting imply current closed-source models tend to have a similar distribution. Evaluation results on more unknown source models can be found in Appendix 8.2.

**Surrogate Model Selection.** Our method can be adapted to any open-source surrogate model. We evaluate our method with different surrogate models including on `GPT-NEO-2.7B`, `Llama2-7B`, and `Llama3-8B`. The experiments are conducted based on Fast-DetectGPT and trained with 5K training data. The results are shown in Table 3 with details in each dataset. In all, compared with the original surrogate models, the surrogate

Table 3: Results comparison of our method with different surrogate models on Claude-3. The performance improvement with our method on different surrogate models shows that our method can be adapted to any open-source surrogate model.

| Surrogate Model | Claude3 PubMed | XSum | Writing |
|---|---|---|---|
| Llama2-7B | 0.8876 | 0.9132 | 0.9243 |
| Llama2-7B(with DALD) | **0.9424** | **0.9773** | **0.9962** |
| Llama3-8B | 0.7764 | 0.9390 | 0.8827 |
| Llama3-8B(with DALD) | **0.9102** | **0.9892** | **0.9967** |
| GPT-Neo-2.7B | 0.8900 | 0.9828 | 0.9445 |
| GPT-Neo-2.7B(with DALD) | **0.8997** | **0.9852** | **0.9515** |

models with `DALD` obtain much higher accuracy. For example, Llama-2 and Llama-3 only obtain 0.8876 and 0.7764 on PubMed-Claude-3 while their counterparts trained with `DALD` achieve 0.9424 and 0.9192. The improvement across various surrogate models suggests that our approach is compatible with a range of surrogates, rather than just a particular carefully chosen surrogate model.

**Ablation Study.** We conduct a group of ablation studies on several datasets, as shown in Table 4. Since our method can be adapted to any previous logits-based methods such as DNA-GPT and Fast-DetectGPT, the ablation study is conducted on top of them to further demonstrate the effectiveness of our method. We choose `Llama2-7B` as the basic surrogate model for all experiments. We compare the results of the baseline model and the baseline model trained with our framework. In general, our method boosts the performance of the baseline model at different scales. For example, DNA-GPT achieves 0.8947 accuracy score on PubMed-GPT-4 while with the surrogate trained by our

Table 4: Ablation study. We report the results comparison of the baseline method and the method with our DALD. The improvement upon all baselines shows the effectiveness of our DALD.

| Method | ChatGPT PubMed | GPT-4 | | | Claude3 | | |
|---|---|---|---|---|---|---|---|
| | | PubMed | XSum | Writing | PubMed | XSum | Writing |
| Detect-GPT | 0.6260 | 0.5291 | 0.6689 | 0.7991 | 0.6472 | 0.9184 | 0.9306 |
| Detect-GPT + DALD | **0.7388** | **0.7034** | **0.8318** | **0.9076** | **0.7550** | **0.9569** | **0.9568** |
| DNA-GPT | 0.9547 | 0.8947 | 0.6980 | 0.8537 | 0.9500 | 0.9359 | 0.9648 |
| DNA-GPT + DALD | **0.9932** | **0.9879** | **0.7524** | **0.9048** | **0.9711** | **0.9391** | **0.9675** |
| Fast-DetectGPT | 0.9309 | 0.8179 | 0.9136 | 0.9521 | 0.8900 | 0.9828 | 0.9445 |
| Fast-DetectGPT + DALD | **0.9853** | **0.9785** | **0.9954** | **0.9980** | **0.9630** | **0.9867** | **0.9981** |

method, DNA-GPT obtains 0.9879 on PubMed-GPT-4. Moreover, we gain a similar conclusion on Fast-DetectGPT. For instance, on PubMed-GPT-4, the original Fast-DetectGPT only has 0.8179 accuracy. However, after training the surrogate model with our method, it achieves 0.9785 accuracy on PubMed-GPT-4. On the one hand, the improvement in baseline shows the effectiveness of our methods. On the other hand, the improvement across different methods demonstrates that our method can be utilized on any logits-based model to boost their performance on black-box detection.

**Non-English Detection.** Recent work[60] finds that current AI detectors are biased for non-English languages, which hinders the application of LLM detection for non-English languages. Following[28], we choose English and German splits of WMT-2016[54] to test the ability of our method in German. We select 150 instances as human-written texts and use the first 30 tokens to regenerate by calling GPT-4 API as machine-generate texts. During training, we randomly select 1K German samples generated by GPT-4 from WildChat[59] and trained for 5 epochs. As shown in Table 5, our method achieves the highest accuracy (> 99%) on German detection compared with DNA-GPT and Fast-DetectGPT. Due to the plug-and-play property, our method can be further used to eliminate the bias in other AI detectors.

Table 5: Results comparison on Non-English texts.

| Method | German-GPT-4 |
|---|---|
| DNA-GPT | 0.7851 |
| Fast-DetectGPT | 0.8814 |
| DALD (Ours) | **0.9955** |

**Adversarial Attack.** In practical situations, machine-generated corpus is often modified and revised by users or another language model. We consider the modified samples as adversarial samples. Evaluating LLM detectors with adversarial samples is important to real-world applications. Following[27] and[28], we randomly mask $r\%$ tokens with 5-word spans in 150 instances from the PubMed dataset regenerated by GPT-4 and apply T5-3B to do the mask-filling task to generate adversarial samples. Experiments with different mask ratios are conducted, specifically $r\% \in \{0.1, 0.2, 0.3, 0.4, 0.5\}$ and results are shown in Figure 6. We compare the results of DNA-GPT, Fast-DetectGPT and their counterparts with our method. In each group, with the enhancement of our method, the model

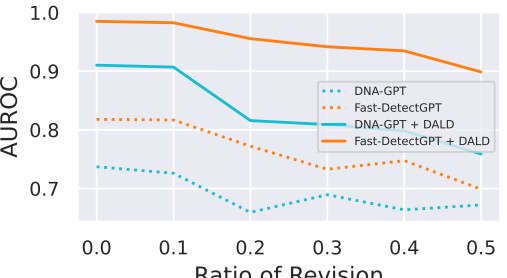

Figure 6: Results comparison on samples with the adversarial attack. The performance improvement with our method on different methods shows that our method is robust to adversarial attacks.

achieves better results on all mask ratios. Moreover, the models with our method obtain even better accuracy at the highest mask ratio compared with the original models on samples without adversarial attack.

**Open-source Model Detection.** In addition to detecting texts from closed-source models, we also evaluate our approach on the open-source model Llama-3, Llama-3.1[61] and Mistral[62] comparing it with DNA-GPT and Fast-DetectGPT. We follow similar settings as closed-source models and fine-tune Llama2-7B as surrogate model. The results are shown in Table 6, where our method works best detection performance on three models compared to other methods, showing the effectiveness of our method on both closed-source and open-source models.

Table 6: Results on open-source models Llama-3, Llama-3.1, and Mistral across three datasets: PubMed (PM), XSum, and Writing. We compare `DALD` with Fast-DetectGPT(Fast).

| Method | Llama3-8B | | | Llama3.1-8B | | | Mistral-7B | | |
| --- | --- | --- | --- | --- | --- | --- | --- | --- | --- |
| | PM | XSum | Writing | PM | XSum | Writing | PM | XSum | Writing |
| Fast | 0.9120 | 0.9845 | 0.9906 | 0.8668 | 0.9914 | 0.9958 | 0.6880 | 0.7931 | 0.9211 |
| `DALD` | **0.9352** | **0.9995** | **0.9972** | **0.9059** | **1.0000** | **0.9998** | **0.7733** | **0.8822** | **0.9573** |

## 5   Conclusion

The rapid evolution of potent Large Language Models (LLMs) underscores the critical necessity for robust black-box detection methods. However, previous methods which rely on surrogate models, suffer from performance degradation, especially with the frequent updates of closed-source models. Our contribution addresses this shortfall by significantly aligning the distribution of the surrogate model and source model. Additionally, we introduce a plug-and-play approach for logits-based detectors, ensuring seamless integration. This method remains versatile across diverse text sources or unknown sources, adapting to the swift evolution of LLMs. In conclusion, our innovations offer compelling solutions to the urgent demand for effective black-box detection methods within the realm of LLM development, bridging critical gaps in current methodologies.

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

# Appendix: DALD

## 6 Theoretical Analysis

### 6.1 Addtional Technical Details

First, we formalize some concepts in preparation of our analysis. We assume the vocabulary set to be $\mathcal{W} = \{w_j\}_{j=1}^{W}$ with size $W$. We will use slices to denote a segment of passage, but unlike python slices, $[a:b]$ will be inclusive on the end points and should denote the indices $\{a, a+1, \cdots, b\}$.

**Text Completion.** Consider the next-token generation process with respect to a language model $f$, which maps an input passage into an array of logits on the vocabulary set $\mathcal{W}$:

**Definition 2.** Given a prompt $X_{[:l_0]}$ of length $l_0$, a new length $l > l_0$ and language model $f$, we use

$$X_{[l_0+1:l]} \sim M_f(X_{[:l_0]}, l)$$

to denote the process of text completion up to a total of $l$ tokens with respect to the following sampling method:

$$X_{l_0+i} \sim \text{softmax}\big(f(X_{[:l_0+i-1]})\big), i = 1, \cdots, l - l_0.$$

In the following we will abbreviate $M_{\text{sur}} = M_{f_{\text{sur}}}$ and $M_{\text{tar}} = M_{f_{\text{tar}}}$, and also denote $p_{\text{sur}}(X) = \text{softmax}\big(f_{\text{sur}}(X)\big)$, $p_{\text{tar}}(X) = \text{softmax}\big(f_{\text{tar}}(X)\big)$ by for any text segment $X$.

**Loss Function for Fine-tuning.** Given target model $f_{\text{tar}}$, we fine-tune a surrogate model $f_{\text{sur}}$ on the following dataset generated by the target model:

$$\mathcal{S} = \big\{X^i = [P^i, R^i]\big|P^i \sim \mathcal{P}, R^i \sim M_{\text{tar}}(P^i, L)\big\}_{i=1}^{N},$$

where each $P^i$ is a prompt sampled from a prior distribution $\mathcal{P}$, and its corresponding $R^i$ is a text completion result sampled from distribution $M_{\text{tar}}$.

Now let us focus on a single text data $X = [P, R] \in \mathcal{S}$ with prompt length $l(P) = l_0$ and total length $l(X) = L$. For any $l_0 < l \le L$, consider the next-token logit $p = p_{\text{sur}}(X_{[:l-1]}) \in [0,1]^T$ of the surrogate model for input $X_{[:l]}$, as well as a sample output from the target model $X_l \sim q = p_{\text{tar}}(X_{[:l-1]})$. The cross-entropy loss is then calculated with

$$\ell(X_l|X_{[:l-1]}) = -\sum_{j=1}^{W} \mathbb{1}\{X_l = w_j\} \log p_j,$$

which has expectation

$$\mathbb{E}_{X_l \sim q}\big[\ell(X_l|X_{[:l-1]})\big] = -\sum_{j=1}^{W} q_j \log p_j = \mathcal{H}(q,p) \ge \mathcal{H}(q),$$

where $\mathcal{H}(q)$ and $\mathcal{H}(q,p)$ denote the entropy of $q$ and the cross-entropy between $q$ and $p$, respectively. It is then evident that the optimal expected value of $\ell(X_l|X_{[:l-1]})$ is $\mathcal{H}(q)$, taken when the surrogate model $f_{\text{sur}}$ produces the same logit output as $f_{\text{tar}}$.

With this, we can express the training objective used for fine-tuning the surrogate model as:

$$\mathcal{L}(f_{\text{sur}}, \mathcal{S}) = \frac{1}{|\mathcal{S}|} \sum_{X=[P,R] \in \mathcal{S}} \sum_{l=l(P)+1}^{l(X)} \ell(X_l|X_{[:l-1]}),$$

## 6.2 Sample Complexity of Fine-tuning

We now make the following assumption, which is supported by many previous works [63]

**Assumption 3.** The sample complexity of fine-tuning with loss $\mathcal{L}(f_{\text{sur}}, \mathcal{S})$ is of order $O(1/\epsilon^2) \times O(\log(1/\delta))$. In other words, when sample size $|\mathcal{S}|$ exceed this order, for almost all $X = [P, R]$ and a corresponding $l > l(P)$, we have

$$\mathbb{E}_{X_l \sim p_{\text{tar}}(X_{[:l-1]})} \ell(X_l | X_{[:l-1]}) \leq \mathcal{H}\big(p_{\text{tar}}(X_{[:l-1]})\big) + \epsilon. \tag{3}$$

**Remark 4.** The result of this assumption is that the surrogate model should approximate the target model well after fine-tuning. We are taking quite a liberty in this assumption in terms of input $X$, but this is simply for the sake of demonstration. In practice, what one can expect is the expected loss $\mathbb{E}_{\mathcal{S}} \mathcal{L}(f_{\text{sur}}, \mathcal{S})$ to be almost optimal, and with generalization this assumption should hold approximately.

**Fast-DetectGPT Method.** For two passages $X, \widetilde{X}$ with the same length $L$, the conditional probability is defined as

$$p_f(\widetilde{X} | X) = \prod_{l=l_0+1}^{L} \mathbb{P}\big[y = \widetilde{X}_l | y \sim \text{softmax}\big(f(X_{[:l-1]})\big)\big],$$

where $l_0$ is a pre-selected length parameter.

In order to evaluate a passage $X$ with length $L$, we sample new text passages $\widetilde{X} \sim p_f(\cdot | X)$ to estimate the conditional probability curvature:

$$\mathbf{d}(X, f) = \frac{\log p_f(X | X) - \widetilde{\mu}}{\widetilde{\sigma}},$$

where

$$\widetilde{\mu} = \mathbb{E}_{\widetilde{X} \sim p_f(\cdot | X)}\big[\log p_f(\widetilde{X} | X)\big], \quad \widetilde{\sigma}^2 = \mathbb{E}_{\widetilde{X} \sim p_f(\cdot | X)}\big[\log p_f(\widetilde{X} | X) - \widetilde{\mu}\big]^2,$$

The basic assumption of Fast-DetectGPT is a positive gap between machine- and human-generated passages in terms of conditional probability curvature:

**Assumption 5.** The conditional probability curvature of a machine-generated passage $X_{\text{tar}}$ from model $f_{\text{tar}}$ is substantially greater than that of a human-generated passage $X_{\text{hum}}$:

$$\mathbf{d}(X_{\text{tar}}, f_{\text{tar}}) - \mathbf{d}(X_{\text{hum}}, f_{\text{tar}}) \geq \Delta.$$

Now with the above assumption, we can prove Theorem 1, which characterizes the difference between the target model and surrogate model.

*Proof of Theorem 1.* A direct result of Assumption 3 is that the Kullback-Liebler devergence between the two distributions are small:

$$\text{KL}\big(p_{\text{tar}}(X_{[:l]}) || p_{\text{sur}}(X_{[:l]})\big) = \mathcal{H}\big(p_{\text{tar}}(X_{[:l]}), p_{\text{sur}}(X_{[:l]})\big) - \mathcal{H}\big(p_{\text{tar}}(X_{[:l]})\big) \leq \epsilon.$$

As per Pinsker's inequality, this gives us

$$d_{\text{TV}}\big(p_{\text{tar}}(X_{[:l]}), p_{\text{sur}}(X_{[:l]})\big) \leq \sqrt{\frac{1}{2}\text{KL}(p_{\text{tar}}(X_{[:l]}) || p_{\text{sur}}(X_{[:l]}))} \leq \sqrt{\frac{\epsilon}{2}}.$$

Now to analyze the difference between conditional probability curvature, we have:

$$\log p_{f_{\text{sur}}}(X | X) - \log p_{f_{\text{tar}}}(X | X) = \sum_{l=l_0+1}^{L} \big[\log p_{\text{sur}}(X_l | X_{[:l-1]}) - \log p_{\text{tar}}(X_l | X_{[:l-1]})\big]$$

$$\leq \sum_{l=l_0+1}^{L} d_{\text{TV}}(p_{\text{sur}}, p_{\text{tar}})/C,$$

where $C = \min p(X_l | X_{[:l-1]})$ is the minimum possibility given to a next-token. For both machine and human generated passages, this value should be reasonably large. On the other hand,

$$
\begin{aligned}
\widetilde{\mu}_{\text{sur}} - \widetilde{\mu}_{\text{tar}} &= \mathbb{E}_{\widetilde{X} \sim p_{f_{\text{sur}}}(\cdot|X)} \log p_{f_{\text{sur}}}(\widetilde{X}|X) - \mathbb{E}_{\widetilde{X} \sim p_{f_{\text{tar}}}(\cdot|X)} \log p_{f_{\text{tar}}}(\widetilde{X}|X) \\
&= \sum_{l=l_0+1}^{L} \left[ \mathbb{E}_{\widetilde{X}_l \sim p_{\text{sur}}(X_{[:l-1]})} \log p_{\text{sur}}(\widetilde{X}_l|X_{[:l-1]}) - \mathbb{E}_{\widetilde{X}_l \sim p_{\text{tar}}(X_{[:l-1]})} \log p_{\text{tar}}(\widetilde{X}_l|X_{[:l-1]}) \right] \\
&= \sum_{l=l_0+1}^{L} \left[ \mathcal{H}(p_{\text{sur}}(X_{[:l-1]})) - \mathcal{H}(p_{\text{tar}}(X_{[:l-1]})) \right] \\
&\leq \sum_{l=l_0+1}^{L} \left[ d_{\text{TV}}(p_{\text{sur}}, p_{\text{tar}}) \log(W-1) + h(d_{\text{TV}}(p_{\text{sur}}, p_{\text{tar}})) \right],
\end{aligned}
$$

where the inequality is from [64]. With this we have that when $\epsilon = O\left[ \left( h^{-1}(\Delta/L) \right)^2 \right]$, the difference $\left| \mathbf{d}(X, f_{\text{sur}}) - \mathbf{d}(X, f_{\text{tar}}) \right| \leq \Delta/3$. This requires the sample complexity to be $\Omega(\text{poly}(\Delta/L))$. $\qquad\square$

An immediate corollary of this theorem is that under the same conditions, $\mathbf{d}(X_{\text{tar}}, f_{\text{sur}}) \geq \mathbf{d}(X_{\text{hum}}, f_{\text{sur}}) + \Delta/3$, which means the positive gap is still present once we replace the target model with the surrogate model.

## 7 More Experimental Details

**Datasets.** We utilize four datasets to evaluate the performance of our method including Xsum[52], PubMedQA[55], WritingPrompts[53] and WTM-2016[54]. Xsum includes documents and corresponding extreme summarization of each news document. PubMedQA is composed of the abstracts of research papers and the corresponding research questions in biomedical research. WritingPrompts is a large dataset consisting of 300K human-written stories paired with writing prompts from an online forum. WTM-2016 is a translation dataset with English-German pairs. For each dataset, we randomly choose 150 examples as human-written texts. Utilizing the 30 prefix tokens of human-written texts, we prompt the target closed-source models by calling API to generate the corresponding texts as the machine-generated text. Some details about prompting for each dataset can be found in Table 7. We utilize the same prompt for different source models including GPT-3.5, GPT-4 and Claude-3. Our training datasets are collected from the open-source datasets, WildChat[59] for GPT-3.5 and GPT-4. Dataset for Claude-3 is generated by calling API with the prompt from WildChat. For GPT-3.5 and GPT-4, we filter the data by timesteps to exactly match the version of the source model and randomly select 5K samples as the training set to fine-tune the surrogate model.

Table 7: Examples of prompts used in different datasets.

| Datasets | Prompts |
|---|---|
| PubMedQA | System: You are a Technical writer. |
| | User: Please answer the question in about 50 words. *Question* |
| Xsum | System: You are a News writer. |
| | User: Please write an article with about 150 words starting exactly with: *Prefix* |
| Writing Prompt | System: You are a Fiction writer. |
| | User: Please write an article with about 150 words starting exactly with:*Prefix* |
| German | System: You are a writer. |
| | User: Please complete a passage in German with about 150 words, starting exactly with: *Prefix* |

**Baselines.** For training-based methods, we compare DALD with GPT-2 detectors[24] developed by OpenAI which build on RoBERTa-base/large[65]. Additionally, we compare DALD to established zero-shot methodologies, such as Likelihood (mean log probabilities), LogRank (average log of ranks in descending order by probabilities), Entropy (mean token entropy of the predictive distribution) [31, 66], and LRR (a fusion of log probability and log-rank) [44].

Table 8: Full results of ChatGPT(GPT-3.5-Turbo) on PubMed, XSum and Writing as the complementary results to Table 1.

| Method | GPT-3.5-Turbo | | |
| --- | --- | --- | --- |
| | PubMed | XSum | Writing |
| DNA-GPT | 0.7788 | 0.9673 | 0.9829 |
| Fast-DetectGPT | 0.9309 | 0.9994 | 0.9967 |
| DALD | 0.9853 | 1.0000 | 1.0000 |

Table 9: More results of our method on one-for-all settings. The surrogate model is trained with 5K samples generated from `GPT-4-0613` and evaluated on other versions of the model including GPT-3.5, GPT-4, GPT-4o and GPT-4o-Mini without further training.

| Method | GPT-3.5-Turbo-0613 PubMed | GPT-3.5-Turbo-1106 PubMed | GPT-3.5-Turbo-0125 PubMed |
| --- | --- | --- | --- |
| DALD | 0.9866 | 0.9911 | 0.9858 |

| Method | GPT-4-1106-Preview | | | GPT-4-0125-Preview | | | GPT-4-0409-Preview | | |
| --- | --- | --- | --- | --- | --- | --- | --- | --- | --- |
| | PubMed | XSum | Writing | PubMed | XSum | Writing | PubMed | XSum | Writing |
| DALD | 0.9780 | 0.9956 | 0.9968 | 0.9850 | 0.9954 | 0.9968 | 0.9815 | 0.9872 | 0.9924 |

| Method | GPT-4o | | | GPT-4o-Mini | | |
| --- | --- | --- | --- | --- | --- | --- |
| | PubMed | XSum | Writing | PubMed | XSum | Writing |
| DALD | 0.9877 | 0.9965 | 0.9994 | 0.9857 | 0.9976 | 0.9992 |

**More Implementation Details.** We describe more implementation details of our framework. For LoRA hyper-parameters, we utilize 16 as the LoRA rank and set lora_alpha as 32. Dropout is set as 0.05. For the Llama series, we adopt the LoRA module on [$"q\_proj"$, $"v\_proj"$, $"k\_proj"$, $"o\_proj"$, $"gate\_proj"$, $"down\_proj"$, $"up\_proj"$] while in GPT-Neo models, we apply on [$"q\_proj"$, $"v\_proj"$, $"k\_proj"$, $"out\_proj"$, $"c\_fc"$, $"c\_proj"$]. For training hyperparameters, we set 512 as the max length for texts from GPT-4 and GPT-3.5 models while it is 2048 for texts from Claude-3. We finetune the surrogate model with a learning rate of 1e-4. The batch size is set as 1 per device with gradient accumulation per 4 steps. It's noteworthy that we do not choose our hyperparameters carefully and we believe it will achieve better results with careful tuning. The training costs less than 10 minutes with 4 A6000.

# 8 Additional Experimental Results

## 8.1 Full Results on ChatGPT(GPT-3.5-Turbo)

We provide the full result on ChatGPT(GPT-3.5-Turbo), as shown in Table 8. Fast-DetectGPT obtains great performance on ChatGPT. Our method boosts their performance with a significant 100% accuracy on Xsum and Writing.

## 8.2 Results on More Versions of Source Model

To show the one-for-all ability of our method, we provide more results on more source models using the surrogate model trained only on data generated from `GPT-4-0613`, as shown in Table 9. We evaluate our method on different versions of GPT-3.5, GPT-4, GPT-4o and GPT-4o-Mini. In general, our method shows robustness on all versions of GPT-3.5 and GPT-4, achieving more than 98% accuracy except on PubMed-GPT-4-1106-Preview but still more than 97%. The superior performance in the one-for-all setting shows the generalizability of our methods, leading to further motivate to train a universal surrogate model for various source models regardless of model updates.

## 8.3 Results on Code Detection

We include a performance comparison of the coding task. We follow [67] to apply APPS[68] dataset as the coding task. We sample 150 coding tasks from APPS and generate the coding results by calling

GPT-4 API. Our method is only trained by the corpus generated by GPT-4 as we previously did. The results are in Table 10, Our method obtains significant improvement on the coding task.

Table 10: Results of code detection on the APPS dataset.

| Method | GPT-4-APPS |
|---|---|
| Fast-DetectGPT | 0.6836 |
| DALD | **0.9078** |

## 8.4 Results on Different Text Genres

We include an evaluation of the detection performance of different text genres. Using a domain-specific datasets RAID, we select 1000 human texts and 1000 GPT-4 generated texts for each domain in RAID dataset and compare the evaluation results with FastDetectGPT. The results is shown in Table 11, which demonstrate that DALD performs admirably in diverse domains.

Table 11: Results across different text genres using the RAID dataset.

| Method | Poetry | News | Abstract | Books | Recipes | Reddit |
|---|---|---|---|---|---|---|
| Fast-DetectGPT | 0.8553 | 0.9116 | 0.8600 | 0.9123 | 0.9116 | 0.9134 |
| DALD | **0.9709** | **0.9567** | **0.9876** | **0.9675** | **0.9998** | **0.9862** |

## 8.5 Results on Different Test Data Sizes

Our main experiment follow previous works such as DNA-GPT and FastDetectGPT and utilize the same amount of data for fair evaluation. Furthermore, we provide the evaluation results of our method on different test data sizes, as shown in Table 12. It is observed that there is little difference in performance as the number of samples increases, indicating the robustness of our method to test data size.

Table 12: Results of ChatGPT and GPT-4 across different sample sizes on PubMed, XSum, and Writing datasets.

| Num. of Samples | ChatGPT PubMed | GPT-4 PubMed | GPT-4 XSum | GPT-4 Writing |
|---|---|---|---|---|
| 150 | 0.9853 | 0.9785 | 0.9954 | 0.9980 |
| 300 | 0.9842 | 0.9821 | 0.9924 | 0.9980 |
| 500 | 0.9806 | 0.9828 | 0.9929 | 0.9974 |

## 8.6 Evaluation Metrics

**AUROC.** Following Fast-DetectGPT[29], we compute the area under the receiver operating characteristic (AUROC) to measure the performance of different methods, evaluating the method performance with different classification thresholds. AUROC falls into the value range $[0, 1]$, which provides a quantitative measure of the likelihood that a randomly generated passage has a higher predicted probability of being machine-generated than a randomly written passage by a human. Generally, the value of 0.5 in AUROC indicates a random classifier without any classification ability while 1 in AUROC refers to a perfect classifier for all samples. The results comparison of our method and other baselines including training-based methods and zero-shot methods are provided in Table 1.

**AUPR.** Similar to AUROC, AUPR computes the area under precision and recall, which evaluates both the precision and recall of classifiers with different thresholds. AUPR also ranges from 0 to 1. A higher AUPR value represents a better classifier, which generally obtains higher recall on the condition that the precision is high. Therefore, we provide the AUPR comparison of our method with other baselines in Table 13. We compare all baselines including training-based such as RoBERTa-base and zero-shot methods such as DetectGPT, DNA-GPT and Fast-DetectGPT. First of all, we can observe that the current mainstream zero-shot methods generally achieve better results compared with the training-based methods. Moreover, our method achieves the highest AUPR results ($> 98\%$) on all datasets and source models, further demonstrating the effectiveness of our method.

Table 13: Performance comparison with baselines on AUPR. Similar to AUROC, our method achieves the best results compared with other training-based methods and zero-shot methods.

| Method | ChatGPT | GPT-4 | | | Claude-3 | | |
| --- | --- | --- | --- | --- | --- | --- | --- |
| | PubMed | PubMed | XSum | Writing | PubMed | XSum | Writing |
| RoBERTa-base | 0.6349 | 0.5538 | 0.7784 | 0.5166 | 0.5325 | 0.8920 | 0.7102 |
| RoBERTa-large | 0.7087 | 0.5887 | 0.7215 | 0.4160 | 0.5466 | 0.8243 | 0.5500 |
| Likelihood | 0.8551 | 0.8453 | 0.9379 | 0.9594 | 0.8453 | 0.9379 | 0.9594 |
| Entropy | 0.3699 | 0.3743 | 0.4172 | 0.3382 | 0.3382 | 0.4172 | 0.3382 |
| LogRank | 0.8811 | 0.8415 | 0.9415 | 0.9504 | 0.9504 | 0.9415 | 0.9504 |
| LRR | 0.7904 | 0.6800 | 0.7370 | 0.7075 | 0.7517 | 0.9070 | 0.8908 |
| NPR | 0.6917 | 0.5793 | 0.6099 | 0.7777 | 0.6124 | 0.8621 | 0.9314 |
| Detect-GPT | 0.5862 | 0.6272 | 0.6940 | 0.6272 | 0.5982 | 0.8551 | 0.9302 |
| DNA-GPT | 0.7181 | 0.6566 | 0.6925 | 0.7610 | 0.6879 | 0.9452 | 0.9540 |
| Fast-DetectGPT | 0.9323 | 0.8315 | 0.9193 | 0.9533 | 0.8955 | 0.9832 | 0.9595 |
| DALD (Ours) | **0.9841** | **0.9802** | **0.9964** | **0.9989** | **0.9881** | **0.9991** | **0.9973** |

# 9 Limitation & Future work

The limitation of our method is the absence of additional experiments conducted in multilingual settings. While our method has shown efficacy in English and German environments, its performance across different languages remains unexplored. Future studies should endeavor to evaluate the generalizability of our approach in multilingual contexts to ensure its applicability across diverse linguistic landscapes.

Table 14: Examples of data from WildChat and Claude-3-Opus-Instruct-15K.

| Datasets | Samples |
|---|---|
| WildChat GPT-3.5-Turbo-0301 | Prompt: who is theseus in plutarch book? |
| | Theseus is a legendary hero and king of Athens in Plutarch's book. Plutarch presents him as a great leader and conqueror who united the scattered towns of Attica and established Athens as the political and cultural center of the region. Theseus is also famous for his defeat of the Minotaur, a monstrous creature with the head of a bull and the body of a man who lived in a labyrinth on the island of Crete. According to Plutarch, Theseus also instituted many reforms in Athens, such as the legal code and the Panathenaic festival. |
| Claude-3-Opus-Instruct-15K | Prompt: Explain how the human circulatory system functions, covering the structures and processes involved in transporting blood throughout the body. Include details about the major components of the system (heart, blood vessels), how blood is pumped through arteries and veins, and the importance of blood circulation for keeping tissues and organs functioning properly. |
| | Response: The human circulatory system is responsible for transporting blood throughout the body in order to deliver oxygen and nutrients to tissues and organs and remove metabolic waste products. The major components are: The Heart: A muscular organ that acts as a pump to propel blood through the blood vessels. It has four chambers - two atria and two ventricles. The right side of the heart pumps deoxygenated blood to the lungs, while the left side pumps oxygenated blood to the rest of the body. Arteries: Blood vessels that carry oxygenated blood away from the heart to the body's tissues. They have thick, muscular walls to withstand the high pressure of blood being pumped from the heart. The aorta is the largest artery. Veins: Blood vessels that carry deoxygenated blood from the tissues back to the heart. They have thinner walls and larger lumens than arteries. Veins have valves to prevent backflow of blood. The venae cavae are the largest veins. Capillaries: Microscopic blood vessels that form a network to facilitate the exchange of substances between the blood and tissues. Oxygen and nutrients diffuse from the blood into cells, while carbon dioxide and wastes move from cells into the blood. The pumping of the heart drives circulation: 1. Deoxygenated blood from the body enters the right atrium, flows into the right ventricle, and is pumped to the lungs via the pulmonary arteries. 2. In the lungs, blood picks up oxygen and releases $CO_2$. Oxygenated blood returns to the left atrium via the pulmonary veins. 3. Blood flows into the left ventricle and is forcefully pumped out to the body through the aorta. 4. The aorta branches into smaller arteries, then arterioles, and finally capillaries where gas and nutrient exchange occurs. 5. Deoxygenated blood collects in venules, flows into veins, and returns to the right atrium via the venae cavae. The cycle repeats. Proper circulation is critical for survival. It ensures that all cells receive oxygen and nutrients needed to function. It enables removal of metabolic wastes that could be toxic if allowed to accumulate. Circulation also helps distribute hormones, transport immune cells, and regulate body temperature and pH. Disruption of blood flow, such as from a blockage, can quickly lead to tissue damage or death. |

