# OpenReview forum: "DALD: Improving Logits-based Detector without Logits from Black-box LLMs"
_NeurIPS.cc/2024/Conference — NeurIPS 2024 poster_

### Official Review · Reviewer_bnRm · 2024-07-11

**Soundness:** 2
**Presentation:** 3
**Contribution:** 3
**Rating:** 5
**Confidence:** 4

**Summary:**

This paper proposes a framework named Distribution-Aligned LLMs Detection (DALD) to improve the performance of surrogate models in detecting LLM-generated text from both closed-source and open-source models. The method enhances detection performance by aligning the distribution of the surrogate model to better match the distribution of the target model. The paper verifies the effectiveness of DALD on various advanced closed-source and open-source models through extensive experiments and demonstrates its "plug-and-play" enhancement capability in a zero-shot detection framework.

**Strengths:**

- It is gratifying that this paper reflects on existing logits-based methods and proposes the DALD framework, which improves detection performance by aligning the distribution of the surrogate model, achieving superior performance.
- The proposed method uses small-scale datasets for fine-tuning, which is cost-effective and highly adaptable, allowing it to quickly adapt to rapid iterations and updates of models, demonstrating good usability.
- Ablation experiments are comprehensive. The framework also shows good performance in scenarios such as Non-English Detection and Adversarial Attack.

**Weaknesses:**

- The method's interpretability is still insufficient. I am not entirely sure, but one point that confuses me is why training DALD on datasets from multiple source models performs better than training on a dataset from a single source model (see Table 2). Since DALD essentially aligns the logits distribution of the surrogate model with that of the source model, this is a crucial starting point of the paper. However, training DALD on datasets from multiple source models theoretically and intuitively would not align distributions better than a single source model, which is contradictory. Similarly, in the Claude-3 setting, DALD trained on a GPT-4 dataset even outperforms DALD trained on a dataset from the source model Claude-3, which is perplexing.
- There is a lack of detail about the datasets used for fine-tuning, such as their sources, topics, or other attributes, which could help readers understand and reproduce the method. In fact, I am very curious about the impact of the fine-tuning dataset on DALD, aside from the sample size.

**Questions:**

- Please refer to the weaknesses, especially Weakness 1.

- An additional question: I noticed that DetectGPT and Fast-DetectGPT are described as black-box methods. Is this because they use surrogate models? In fact, all logits-based methods (including Likelihood and Rank) can use surrogate models, depending on whether the scenario is white-box or black-box.

**Limitations:**

- The testing dataset is insufficient; all datasets contain only 150 human-written samples, and the experimental results obtained may actually be biased, which is indeed my concern.

---

> ### Author Rebuttal · Authors · 2024-08-06
>
> Thank you very much for your detailed review. Please check our response to your concerns.
>
> **Response to Weakness 1**:
>
> First, regarding the concern of the better performance of the model trained on multiple source data, a reasonable explanation could be that the optimization space of logits is large. Therefore, aligning the logits of one model does not necessarily affect the alignment of others in the LLM detection point of view. It suggests that continuing to learn from the corpus generated by multiple models simultaneously has no negative impact on the overall performance. Moreover, by collecting from multiple sources, the training data is augmented, which, given homogeneous models, is similar in spirit to the widely used data augmentation for improving generalization.  Augmentations induce extra stochasticity during training, which effectively flattens the loss landscape [3]. The surrogate model may also benefit from the complementary information from different sources that can potentially improve the alignment of distributions, and eventually enhance the model's generalization performance. We will add discussions in the revision. This indicates that training DALD on datasets from multiple source models is indeed effective and not contradictory.
>
> Besides, regarding the point you raised about the perplexing outperformance of DALD trained on a GPT-4 dataset compared to a dataset from the source model Claude3, we would like to point out that their performance is close. In the general direction, a closer distribution of logits can help produce better detection results. However, there is a certain degree of randomness involved, as we are not comparing the direct value of logits, but rather the scoring function in Fast-DetectGPT, which uses logits to calculate the score as the detection metric. In most cases, our results align with our assumptions.
>
> **Response to Weakness 2**:
>
> Sorry for the unclearness of our fine-tuning datasets. As we discussed in the main text, the only requirement of the fine-tuning dataset is that the texts are generated by the same target model. More concretely, in our experiments, we utilize the corpus generated by ChatGPT and GPT-4 from WildChat[1]. As for Claude-3, the data is from the huggingface[2]. For the open-source model, we randomly select 5000 prompts from Wildchat and generate the output from the open-source models to obtain fine-tuning data. We will share our code with the public for reproduction.
>
> **Response to Question 2**:
>
> DetectGPT and FastDetectGPT can be applied in both white-box and black-box settings. In the white-box setting, the output logits of the target model will be accessed. However, it is unavailable in the black-box setting, thus methods like DetectGPT and FastDetectGPT utilize a surrogate model to compute the output logits. Our method focuses on the black-box setting since the black-box scenario is more practical but challenging due to the closed-source trend of current models. Therefore, we report and compare the results of DetectGPT and FastDetectGPT in the black-box setting.
>
> **Response to Limitation**:
>
> Thank you for your feedback on the size of the test dataset. First of all, we follow previous works such as  DNA-GPT and FastDetectGPT and utilize the same amount of data for fair evaluation. Moreover, we are happy to provide the evaluation results of our method on different test data sizes, as shown:
>
> |Num. of Samples|\||ChatGPT|\|||GPT4||
> |---|---|---|---|---|---|---|
> ||\||Pubmed|\||Pubmed|Xsum|Writing|
> |150|\||0.9853|\||0.9785|0.9954|0.9980|
> |300|\||0.9842|\||0.9821|0.9924|0.9980|
> |500|\||0.9806|\||0.9828|0.9929|0.9974|
>
> It is observed that there is little difference in performance as the number of samples increases, indicating the robustness of our method to test data size.
>
> **Reference**:
>
> [1] Zhao, Wenting, et al. "Wildchat: 1m chatGPT interaction logs in the wild." arXiv preprint arXiv:2405.01470 (2024).
>
> [2] lodrick-the-lafted/Sao10K_Claude-3-Opus-Instruct-9.5K-ShareGPT
>
> [3] Geiping, Jonas, et al. "How much data are augmentations worth? An investigation into scaling laws, invariance, and implicit regularization." arXiv preprint arXiv:2210.06441 (2022).

---

> > ### Comment · Reviewer_bnRm · 2024-08-13
> > **Official Comment by Reviewer bnRm**
> >
> > I would like to thank the responses provided by authors. However, my concerns about the Weakness 1 are not well addressed. Specifically, regarding the observation that "DALD trained on the GPT-4 dataset performs better than the source model Claude3's dataset," are the logits distributions of GPT-4 and Claude3 truly similar? This contradicts my experience, and I still have doubts about the consistency between the assumptions and experimental results.

---

> > > ### Author Response · Authors · 2024-08-13
> > > **Response to Reviewer bnRm**
> > >
> > > Dear Reviewer bnRm,
> > >
> > > Thank you very much for your reply. We really appreciate your suggestions on our work. Regarding your concern about our assumption and experimental results, as we discussed in the main text, the training data generated by GPT-4 is from WildChat[1], which is a high-quality dataset. However, WildChat does not include the data generated from Claude3. In order to save money, we didn't call Claude3 API to generate training data. The data of Claude3 was collected from the public output in Huggingface without a quality guarantee[2]. Therefore, the performance of the model trained by data generated from Claude is relatively worse than that trained by data generated from GPT-4.
> > >
> > > For a fair comparison, we sampled 5k prompts from wildchat and called Claude API to generate the training dataset. Then, we utilize the generated dataset to fine-tune the surrogate model and provide the results here(we will also include this in the next version):
> > >
> > > |                     | PubMed | Xsum   | Writing |
> > > |---------------------|--------|--------|---------|
> > > | DALD(GPT-4 data)    | 0.9875 | 0.9993 | 0.9977  |
> > > | DALD(Claude-3 data) | 0.9942 | 0.9994 | 0.9993  |
> > >
> > > The results show that the model trained by data generated from Claude3 is better than the one trained by data from GPT-4, demonstrating the correctness of our assumption and the consistency of experimental results.
> > >
> > >
> > > [1] Zhao, Wenting, et al. "Wildchat: 1m chatGPT interaction logs in the wild." arXiv preprint arXiv:2405.01470 (2024).
> > >
> > > [2] lodrick-the-lafted/Sao10K_Claude-3-Opus-Instruct-9.5K-ShareGPT

---

### Official Review · Reviewer_3RMB · 2024-07-12

**Soundness:** 3
**Presentation:** 3
**Contribution:** 3
**Rating:** 8
**Confidence:** 3

**Summary:**

This paper proposes a method to improve black-box detection of machine-generated text, tackling the problem of performance degradation when a surrogate model's output is poorly aligned with the closed-source target LLM. By LORA fine-tuning the surrogate model on text generated by the target model, the authors align the surrogate model's output distribution to the target model, thus improving detection performance.

The authors show that this alignment can be used to outperform/improve existing logit detection methods such as DetectGPT, DNA-GPT and Fast-DetectGPT; is effective across a range of surrogate models; and requires only a small amount of fine-tuning (and thus can be keep surrogate models up-to-date when newer models are released).

**Strengths:**

- **Consistent performance improvements**: Ablation study (Table 4) suggests that DALD leads to consistent and significant improvements across target models and datasets, for all three logit-based methods
 - **Surprisingly effective**: it's quite surprising to me that a relatively low amount of data/fine-tuning is so effective at aligning the probability distributions of target and surrogate models. I wonder how effective DALD is for more niche tasks/text genres, in cases where there may be less overlap between the training corpus and the evaluation text? (the authors briefly touch on this when discussing PubMed results)
- **Application to non-English & practical settings**: The authors demonstrate that DALD can also help tackle the issue of non-English detection bias by showing performance improvements for German. They also show that their method is robust to machine-generated text that has undergone adversarial attacks.

**Weaknesses:**

- **Generalizability may be overstated** (Table 2): The authors cite the superior performance of their one-for-all surrogate model as evidence that their method could be extended to detect texts of unknown source models. However, the improvements seems very slight: I agree with their interpretation that current closed-source models tend to have a similar distribution, but this homogeneity might not necessarily be true for future versions of these models and/or other future closed-source models.
- **Effectiveness across surrogate models** (Table 3): DALD doesn't significantly improve the detection performance of all surrogate models. For example, its effect on GPT2-Neo 2.7B, though positive, seems fairly minor. Does this indicate a limitation of their DALD method (e.g. that it is only effective when the performance gap/parameter size/pre-training data difference between surrogate and target is relatively small)?
- **Limitation and Future work section** is extremely brief, and does not adequately address limitations

**Questions:**

See Weaknesses Section. (and Strengths bullet point 2)

Also, minor nitpicks / typos:


Typos:
- Line 187: is freezing
- Line 215: OpanAI
- Line 244: pertraining
- Line 574: We

**Limitations:**

Limitations section was extremely brief and only discussed extending their evaluation of multilingual performance.

---

> ### Author Rebuttal · Authors · 2024-08-06
>
> Thank you very much for the constructive and detailed reviews. We provide detailed responses to your concerns.
>
> **Response to the question about different domains**:
>
> We appreciate your interest in the effectiveness of DALD for different domains. It's worth noting that our training data is entirely distinct from our test dataset, so there is no overlap between the two datasets. Training data is the corpus from the publicly shared outputs of leading models while testing data is the specific datasets such as PubMed prompted by the target models. Additionally, we've conducted experiments on other domain-specific datasets, namely RAID[1], including poetry, news, books, etc. Results are shown:
>
> ||\||Poetry|News|Abstract|Books|Recipes|Reddit|
> |---|---|---|---|---|---|---|---|
> |**Fast-DetectGPT**|\||0.8553|0.9116|0.8600|0.9123|0.9116|0.9134|
> |**DALD**|\||0.9709|0.9567|0.9876|0.9675|0.9998|0.9862|
>
> We select 1000 human texts and 1000 GPT-4 generated texts for each domain in RAID dataset and compare the evaluation results with FastDetectGPT. The results demonstrate that DALD performs admirably in diverse domains.
>
> **Response to Weakness 1**:
>
> Thank you for raising this point as it provides a deep insight into our method. We acknowledge empirically that current closed-source models have homogeneity characteristics, as evidenced by the slight improvement in multisource settings. Therefore, we cannot guarantee the one-for-all surrogate model can perform optimally when encountering new models without homogeneity. However, it's important to note that the generalizability of our approach also lies in its ability to continuous learning. By collecting small-size data from new models and continually fine-tuning the surrogate model, we can adapt to newly published models with minimal cost. This ongoing effort represents a worthwhile alternative compared to retraining the entire model from scratch.
>
> **Response to Weakness 2**:
>
> We appreciate this careful review. The modest performance improvement of GPT-Neo-2.7B as the surrogate model only happens on Claude3 detection. As shown in the table below, upon further testing on GPT-4, it is observed a substantial improvement in its performance. We believe this might be attributed to that the parameter number of GPT-Neo-2.7B is relatively small and may not be sufficient to generalize well across all datasets. We intend to present a comprehensive overview of GPT-Neo-2.7B in the next version.
>
> ||\||Pubmed|Xsum|Writing|
> |---|---|---|---|---|
> |**Fast-DetectGPT**|\||0.8179|0.8179|0.9521|
> |**DALD(GPT-Neo-2.7B)**|\||0.9020|0.9732|0.9800|
>
>
> **Response to Weakness 3**:
>
> We acknowledge that the Limitation and Future Work section is brief since we have to condense this section in the current version due to space constraints. However, we appreciate your feedback and will ensure to include a more comprehensive discussiont in the next version of the paper. Regarding future work, we plan to explore the influence function to improve data efficiency for finetuning. Additionally, future work can utilize our pipeline for analyzing the homogeneity of different models and versions such as investigating whether there is more variability in non-transformer architectures or black-box model alignment from different companies. These are crucial areas that we will focus on exploring further.
>
>
> **Response to Typo**:
>
> Thank you for this careful review and pointing out the typo in our paper. We appreciate your attention to detail and will make sure to revise the typo accordingly in the next version of the manuscript. Your feedback is invaluable, and we are grateful for your thorough assessment of our work!
>
> **Reference**:
>
> [1] Dugan, Liam, et al. "RAID: A Shared Benchmark for Robust Evaluation of Machine-Generated Text Detectors." arXiv preprint arXiv:2405.07940 (2024).

---

> > ### Comment · Reviewer_3RMB · 2024-08-12
> >
> > Thanks for your response. I appreciate the additional results you've provided on domain-specific datasets as well as for GPT-Neo-2.7B on GPT-4 (Weakness 2).

---

> > > ### Author Response · Authors · 2024-08-12
> > > **Response to Reviewer 3RMB**
> > >
> > > Dear Reviewer 3RMB,
> > >
> > > Thank you very much for your comments. We will revise our manuscript accordingly in the next version.
> > >
> > > Best,
> > >
> > > Author

---

### Official Review · Reviewer_yZjY · 2024-07-12

**Soundness:** 2
**Presentation:** 3
**Contribution:** 2
**Rating:** 5
**Confidence:** 3

**Summary:**

This paper introduces Distribution-Aligned LLMs Detection (DALD), a novel framework for detecting AI-generated text from large language models (LLMs). DALD addresses limitations of traditional detection methods, particularly when dealing with black-box or unknown LLMs. It aligns surrogate models with unknown target LLM distributions, enhancing detection capability across various models without requiring access to their internal logits. The approach leverages corpus samples from publicly available outputs of advanced models to fine-tune surrogate models, achieving state-of-the-art performance on both closed-source and open-source models. DALD can be integrated into existing zero-shot detection frameworks and demonstrates robustness against revised text attacks and non-English texts, offering a versatile solution for the evolving challenge of distinguishing AI-generated content from human-written text.

**Strengths:**

1. The paper proposes a highly effective text detection framework.
2. The paper utilizes the fact that models can generate text under black-box settings to train a surrogate model, achieving efficient prediction in black-box scenarios.
3. The framework presented in the paper is actually plug-and-play and can be integrated into logit-based detection methods such as DetectGPT and Fast-DetectGPT.

**Weaknesses:**

1. Since DALO is a training-based method, it may be unfair to compare it with the zero-shot method(Detect-GPT, DNA-GPT, and Fast-DetectGPT). Comparing it with other SOTA training-based methods such as Ghostbuster[1] is necessary.
2. Only testing PubMed on ChatGPT while testing all three datasets on GPT-4 and Claude.
3. For reliability, it is important to see how well the detector performs at low FPR regimes [2].
4. Expanding your evaluation to include recent open-source models such as Llama 3 and Mistral/Mixtral would enhance the breadth and relevance of your study.
5. The method is easy and lacks contribution in technology.

[1] Vivek Verma, Eve Fleisig, et al. "Ghostbuster: Detecting text ghostwritten by large language models" (2023).
[2] Sadasivan, Vinu Sankar, et al. "Can ai-generated text be reliably detected?" arXiv preprint arXiv:2303.11156 (2023).

**Questions:**

1. Could you compare it with other SOTA training-based methods such as Ghostbuster[1]?
2. Why did you just test PubMed on ChatGPT while testing all three datasets on GPT-4 and Claude?
3. Could you provide the ROC curves for your detector? For reliability, it is important to see how well the detector performs at low FPR regimes [2].
4. Why not conduct some experiments on open-source LLM like Llama 3 and Mistral/Mixtral？

[1] Vivek Verma, Eve Fleisig, et al. "Ghostbuster: Detecting text ghostwritten by large language models" (2023).
[2] Sadasivan, Vinu Sankar, et al. "Can ai-generated text be reliably detected?" arXiv preprint arXiv:2303.11156 (2023).

**Limitations:**

Yes

---

> ### Author Rebuttal · Authors · 2024-08-06
>
> Thanks a lot for your detailed and careful reviews. We will give our response to your questions.
>
> **Response to Weakness 1**:
>
> We appreciate the suggestion of comparison with Ghostbuster and adding experiments comparing our method to Ghostbuster. We utilize the official code of Ghostbuster and evaluate it on the same datasets in our experiments. The results are:
>
> ||\||GPT-3.5|\|||GPT-4||\|||Claude3||
> |---|---|---|---|---|---|---|---|---|---|---|
> ||\||Pubmed|\||Pubmed|Xsum|Writing|\||Pubmed|Xsum|Writing|
> |**GhostBuster**|\||0.8108|\||0.7269|0.8384|0.9614|\||0.7722|0.946|0.9675|
> |**DALD**|\||0.9853|\||0.9785|0.9954|0.9980|\||0.9630|0.9867|0.9981|
>
> DALD achieves the best performance on all datasets compared with Ghostbuster. We will add the results of Ghostbuster and more discussions to the next version. In addition, we would like to point out that we have already compared our method to training-based methods like Roberta (we gently invite the reviewer to check Table 1), demonstrating its effectiveness in comparison to existing approaches. Concerning the fairness of comparison, we would like to clarify that our method significantly differs from traditional training-based methods, as we only utilize a small amount of data (e.g., around 2k data) to fine-tune the model (less than 10 mins on a single A6000), whereas training-based methods typically require much more training data (250K in Roberta). Therefore, we believe that to demonstrate the uniqueness and advantages of our method, it is essential (and has been) to compare it with both zero-shot and training-based methods.
>
> **Response to Weakness 2**:
>
> We appreciate your careful review. Due to space constraints in the paper, we omitted results from two other datasets on ChatGPT, so we had to prioritize the study using the most recent datasets available on PubMed.  We add the results here (which will also be included in the next version):
>
> ||ChatGPT-Pubmed|ChatGPT-Xsum|ChatGPT-Writing|
> |---|---|---|---|
> |**DNA-GPT**|0.7788|0.9673|0.9829|
> |**Fast-DetectGPT**|0.9309|0.9994|0.9967|
> |**GhostBuster**|0.8108|0.9832|0.9983|
> |**DALD**|0.9853|1.0000|1.0000|
>
> It is observed Xsum and Writing on ChatGPT can be well detected by FastDetectGPT. Furthermore, our method further improves the performance on top of FastDetectGPT.
>
>
> **Response to Weakness 3**:
>
> Thank you for this suggestion. We will add the ROC curves to the pdf page. We would like to invite you to check Figure 1 on the pdf page. We compare the curve with DNA-GPT and FastDetectGPT. Our method achieves the best performance at low FPR on all datasets.
>
> **Response to Weakness 4**:
>
> We appreciate the suggestion to incorporate recent open-source models such as Llama 3 and Mistral in our assessment. It's worth noting that the results for Llama 3-8B are already documented in Table 6. Additionally, we intend to present further findings on open-source models like Mistral-7B and the new Llama 3.1-8B. These results are available for review:
> ||\||LLama3.1|||\||Mistral|||
> |---|---|---|---|---|---|---|---|---|
> ||\||Pubmed|Xsum|Writing|\||Pubmed|Xsum|Writing|
> |**Fast-DetectGPT**|\||0.8668|0.9914|0.9958|\||0.6880|0.7931|0.9211|
> |**DALD**|\||0.9059|1.0000|0.9998|\||0.7733|0.8822|0.9573|
>
> The results show that our method achieves the best on all open-source models, demonstrating its effectiveness.
>
> **Response to Weakness 5**:
>
> While our method might be easy to implement, it's essential to consider the practical implications and benefits it offers. We want to highlight that our work's most significant contribution is the observation that lightweight distribution alignment (less than 10 mins on a single A6000) between the surrogate model and the target model can significantly improve detection accuracy and effectively address the issue of model updates. The empirical results demonstrate the strong sample efficiency of the proposed approach, which is also supported by theoretical analysis. Furthermore, the empirical results reveal an interesting observation: the approach exhibits relatively strong generalization capabilities and can be used to enhance all existing logit-based detectors.

---

> > ### Comment · Reviewer_yZjY · 2024-08-13
> >
> > I'm glad for your comprehensive clarification. I'm curious about what you mentioned regarding space constraints in weakness 2, as I recall that the appendix isn't page-limited. Besides, I agree with the reviewer bnRm's weakness 1 about training on different domains. I will keep my score.

---

> > > ### Author Response · Authors · 2024-08-13
> > > **Response to Reviewer yZjY**
> > >
> > > Dear Reviewer yZjY,
> > >
> > > Thank you very much for your advice on our work, which greatly enhances the solidity of our work. We will add all related experimental results to our main text in the next version.
> > >
> > > Regarding your concern about our assumption and experimental results, as we discussed in the main text, the training data generated by GPT-4 is from WildChat[1], which is a high-quality dataset. However, WildChat does not include the data generated from Claude3. In order to save money, we didn't call Claude3 API to generate training data. The data of Claude3 was collected from the public output in Huggingface without a quality guarantee[2]. Therefore, the performance of the model trained by data generated from Claude is relatively worse than that trained by data generated from GPT-4.
> > >
> > > For a fair comparison, we sampled 5k prompts from wildchat and called Claude API to generate the training dataset. Then, we utilize the generated dataset to fine-tune the surrogate model and provide the results here(we will also include this in the next version):
> > >
> > >
> > > |                     | PubMed | Xsum   | Writing |
> > > |---------------------|--------|--------|---------|
> > > | DALD(GPT-4 data)    | 0.9875 | 0.9993 | 0.9977  |
> > > | DALD(Claude-3 data) | 0.9942 | 0.9994 | 0.9993  |
> > >
> > > The results show that the model trained by data generated from Claude3 is better than the one trained by data from GPT-4, demonstrating the correctness of our assumption and the consistency of experimental results.
> > >
> > >
> > > [1] Zhao, Wenting, et al. "Wildchat: 1m chatGPT interaction logs in the wild." arXiv preprint arXiv:2405.01470 (2024).
> > >
> > > [2] lodrick-the-lafted/Sao10K_Claude-3-Opus-Instruct-9.5K-ShareGPT

---

### Official Review · Reviewer_iaeF · 2024-07-13

**Soundness:** 3
**Presentation:** 2
**Contribution:** 2
**Rating:** 4
**Confidence:** 3

**Summary:**

The paper addresses the challenge of detecting machine-generated text from black-box LLMs without access to their logits. Traditional methods using surrogate models suffer from performance degradation due to misalignment with target model distributions, particularly as new models are introduced. The proposed Distribution-Aligned LLMs Detection (DALD) framework aligns surrogate model distributions with unknown target LLMs. DALD enhances detection capabilities, achieving state-of-the-art performance in black-box settings, and offers a plug-and-play enhancement to existing zero-shot detection frameworks. Extensive experiments validate DALD's high detection precision against revised text attacks and non-English texts.

**Strengths:**

1. The detection of machine-generated text is an important problem.
2. The paper includes theoretical analysis on the effectiveness of fine-tuning.

**Weaknesses:**

1. The idea lacks novelty; collecting data from models with the same version as the target model and then aligning the surrogate model is not novel.
2. The proposed method appears to heavily rely on alignment data collection and specific model versions. Does this mean the surrogate model needs continuous fine-tuning, when the detection task varies? If so, what is the generalizability?
3. The paper missing abalation studies on how much fine-tuning data is required when the detection task varies in different categories. If the data requirement is large, it could pose a problem for the detection task.
4. The paper is missing important baselines such as RADAR [1] and could benefit from evaluation on more models like Gemini and additional tasks, including coding tasks.

## Reference

[1]. Radar: Robust ai-text detection via adversarial learning. NeurIPS 2023.

**Questions:**

Please refer to the weaknesses.

**Limitations:**

Please refer to the weaknesses.

---

> ### Author Rebuttal · Authors · 2024-08-06
>
> Thank you very much for your detailed reviews. Here are our responses to your questions:
>
> **Response to Weakness 1**:
>
> The detection of black-box commercial LLM models is a critical topic. The rapid updates in these models present a significant and ongoing challenge, as they lead to decreased detection efficiency in existing logit-based zero-shot detectors. Traditional methods require the manual and careful selection of different surrogate models. Our work's most significant contribution is the observation that lightweight distribution alignment (less than 10 mins on a single A6000) between the surrogate model and the target model can significantly improve detection accuracy and effectively address the issue of model updates. The empirical results demonstrate the strong sample efficiency of the proposed approach, which is also supported by theoretical analysis. Furthermore, the empirical results reveal an interesting observation: the approach exhibits relatively strong generalization capabilities and can be used to enhance all existing logit-based detectors.
>
> **Response to Weakness 2**:
>
> There are two aspects of the generalizability of our method.
> On the one hand, the model trained on single-source data can perform effectively on different target models, which indicates one-for-all ability. We gently invite the reviewer to check the 3rd line results in Table 2. Our model is only trained on 5K texts generated by GPT-4, yet it demonstrated large performance improvement on ChatGPT, GPT-4, and Claude3.  Additionally, we further tested its performance on the newly published models GPT-4o and GPT-4o-mini and found the model also has strong performance on them. These results highlight the generalizability of our approach.
>
>
> ||\||GPT-4|||\||Claude3|||\||GPT-4o|||\||GPT-4o-Mini|||
> |---|---|---|---|---|---|:---------:|---|---|---|---|---|---|---|---|---|---|
> ||\|| PubMed|Xsum|Writing| \||PubMed|Xsum|Writing |\|| PubMed|Xsum| Writing|\|| PubMed | Xsum| Writing|
> | **Baseline**| \| |0.7995|0.7072|0.9299|\|| 0.8877|0.9143| 0.9248|\||0.8163|0.6595| 0.9505|\|| 0.7994 | 0.5877| 0.9150|
> | **DALD** | \|| 0.9785 | 0.9954 | 0.998 |\|| 0.9875 | 0.9993| 0.9977  |\|| 0.9877 | 0.9965 | 0.9994  |\|| 0.9857 | 0.9976 | 0.9992|
>
>
> On the other hand, reviewer bnrm also highlighted that the model has the capability of continuous learning. We encourage the reviewer to refer to the 5th line of Table 2 where we present the results of our model which has been trained using 5K texts from multiple sources. Its exceptional performance suggests that if a new closed-source model is released, we can gather a small corpus generated by the model and use it to continue training the surrogate model. Thus, our model can rapidly and continuously adapt to new models with minimal fine-tuning costs.
>
> **Response to Weakness 3**:
>
> First, our paper contains in-depth experiments that investigate the required data size for fine-tuning. We would like to invite the reviewer to refer to Figure 4, which depicts the performance at various data sizes. This result shows that just about 2k of training data is enough to yield significant improvements, indicating the minimal amount of data needed. Moreover, we performed a theoretical analysis of the amount of training data (refer to the Appendix and the Method section). One of our key contributions is demonstrating that only a minimal tuning cost is required.
>
> **Response to Weakness 4**:
>
> Thank you for your suggestions about adding more baseline models and different tasks. As suggested, we add comparisons with RADAR:
>
> ||\||GPT-3.5|\|||GPT-4||\|||Claude3||
> |---|---|---|---|---|---|---|---|---|---|---|
> ||\||Pubmed|\||Pubmed|Xsum|Writing|\||Pubmed|Xsum|Writing|
> |**RADAR**|\||0.8953|\||0.8818|0.9926|0.8496|\||0.8295|0.9900|0.8780|
> |**DALD**|\||0.9853|\||0.9785|0.9954|0.9980|\||0.9630|0.9867|0.9981|
>
>
> We utilize the official codebase of RADAR. RADAR is evaluated on the same test dataset as our DALD. As shown in the table, DALD achieves better performance on all datasets. We will add the comparison and discussion to our draft accordingly.
>
> We also include the performance comparison on the coding task. We follow [2] to apply APPS[1] dataset as the coding task. We sample 150 coding tasks from APPS and generate the coding results by calling GPT-4 API. Our method is only trained by the corpus generated by GPT-4 as we previously did. The results are as follows:
>
> ||\||GPT4-APPS|
> |---|---|:---:|
> | **RADAR**| \| | 0.5067|
> | **Fast-DetectGPT** | \| | 0.6836|
> | **DALD** | \| | 0.9078|
>
> Our method obtains significant improvement on the coding task.
>
> Finally, we conduct the experiments on Gemini, as shown in:
> ||\||Gemini-PubMed|
> |---|---|:---:|
> | **RADAR**| \| |0.8496|
> | **Fast-DetectGPT** | \| | 0.9352|
> | **DALD** | \| | 0.9902|
>
> Since Gemini has the protection mechanisms(recitation), we only conducted the experiments on the PubMed dataset to evaluate the performance. Our method is still the best compared with other methods, demonstrating the effectiveness of our method.
>
>
> **Reference**:
>
> [1] Hendrycks, Dan, et al. "Measuring coding challenge competence with apps." arXiv preprint arXiv:2105.09938 (2021)
>
> [2] Yang, Xianjun, et al. "Zero-shot detection of machine-generated codes." arXiv preprint arXiv:2310.05103 (2023).

---

> > ### Author Response · Authors · 2024-08-13
> > **Reply to Reviewer iaeF**
> >
> > Dear Reviewer iaeF,
> >
> > This is a kind reminder. We wanted to kindly follow up on the rebuttal discussion regarding. We highly value your insights and would greatly appreciate your response at your earliest convenience. Please let me know if there is any additional information you require or if there are any concerns we can address.
> >
> > Thank you for your time and consideration.
> >
> > Authors

---

### Author Rebuttal · Authors · 2024-08-06

**PDF Pages**:

This PDF page includes the ROC curves comparison with DNA-GPT and Fast-DetectGPT. We would like to invite all reviewers to check the Figure in the page. Thanks a lot.

---

### Author Response · Authors · 2024-08-07
**General Response**

## General Response

Dear AC and Reviewers,

We would like to sincerely appreciate the reviewers for their positive feedback and highly constructive comments. To improve the clarity and readability of the paper, the following changes have been made and the manuscript will be revised accordingly in the next version.

* We clarified some misunderstandings of the reviewers about the method and experiment.
* We added more experiments and explanations to demonstrate the generalizability of DALD.
* We added more baseline comparisons including Ghostbuster and RADAR.
* We provided the ROC curves of our detector and other methods.
* We added more experiments on open-source models including llama3.1-8B, and Mistral-7B.
* We added more experiments on RAID datasets with more topics.
* Add a detailed discussion of the limitations and future work.

Thanks again,

The Authors

---

### Decision · Program_Chairs · 2024-09-25

**Decision:**

Accept (poster)

**Comment:**

The paper proposes a method to fine-tune a surrogate LLM against texts generated by the target LLM as a means of improving the ability to detect/guess if a given text is generated from the aforementioned target LLM. The idea is simple and requires a relatively small number of generations to improve the model's efficacy. The reported effect size is quite significant, by 10% in many cases.

While a number of borderline reviews are present, most of the concerns are non-falsifiable. e.g., lacking novelty but not citing prior work that has followed this approach, or that the explanation for the idea/why it works is not sufficient (what makes something sufficient?). I do not consider these substantive reasons to reject the work, given that the effect size is large (let theory follow empirical results) and the simplicity of the approach (if it does not work in other contexts, it will not be hard to find out).

For these reasons, I am recommending the paper for the proceedings of the conference.